# Functional MRI Assessment of Brain Activity Patterns Associated with Reading in Medulloblastoma Survivors

**DOI:** 10.3390/brainsci14090904

**Published:** 2024-09-06

**Authors:** Josue L. Dalboni da Rocha, Ping Zou Stinnett, Matthew A. Scoggins, Samuel S. McAfee, Heather M. Conklin, Amar Gajjar, Ranganatha Sitaram

**Affiliations:** 1Department of Diagnostic Imaging, St. Jude Children Research’s Hospital, Memphis, TN 38105, USA; josueluiz.dalbonidarocha@stjude.org (J.L.D.d.R.); ping.zou@stjude.org (P.Z.S.); stuart.mcafee@stjude.org (S.S.M.); 2Department of Psychology and Biobehavioral Sciences, St. Jude Children Research’s Hospital, Memphis, TN 38105, USA; matthew.scoggins@stjude.org (M.A.S.); heather.conklin@stjude.org (H.M.C.); 3Department of Pediatric Medicine, St. Jude Children Research’s Hospital, Memphis, TN 38105, USA; amar.gajjar@stjude.org

**Keywords:** medulloblastoma, cognitive impairment, fMRI, rapid automatized naming, reading, brain tumor

## Abstract

Medulloblastoma, a malignant brain tumor primarily affecting children, poses significant challenges to patients and clinicians due to its complex treatment and potential long-term cognitive consequences. While recent advancements in treatment have significantly improved survival rates, survivors often face cognitive impairments, particularly in reading, impacting their quality of life. According to the double deficit theory, reading impairments are caused by deficits in one or both of two independent reading-related functions: phonological awareness and rapid visual naming. This longitudinal study investigates neurofunctional changes related to reading in medulloblastoma survivors in comparison to controls using functional MRI acquired during rapid automatized naming tasks over three annual visits. Support vector machine classification of functional MRI data reveals a progressive divergence in brain activity patterns between medulloblastoma survivors and healthy controls over time, suggesting delayed effects of cancer treatment on brain function. Alterations in brain regions involved in visual processing and orthographic recognition during rapid naming tasks imply disruptions in the ventral visual pathway associated with normal orthographic processing. These alterations are correlated with performance in tasks involving sound awareness, reading fluency, and word attack. These findings underscore the dynamic nature of post-treatment neurofunctional alterations and the importance of early identification and intervention to address cognitive deficits in survivors.

## 1. Introduction

Medulloblastoma is the most common malignant brain tumor in children, accounting for approximately 20% of all pediatric brain tumors [1]. It predominantly affects the pediatric population, with a 10-fold higher frequency than in adults [2], particularly in children between the ages of 2 and 8 years [3]. Medulloblastoma primarily arises in the cerebellum or posterior brainstem located within the posterior fossa [4]. The cerebellum is known to interact with supratentorial somatomotor circuits to coordinate movement, and increasing evidence suggests it also plays a role in coordinating cognitive processes with the cerebrum [5,6]. Clinically, medulloblastoma often presents with symptoms related to increased intracranial pressure, such as headaches, nausea, vomiting, and balance difficulties, due to its location in the posterior fossa [1]. In younger children, irritability and developmental delay may also be observed. Treatment for medulloblastoma includes maximal surgical resection, radiation therapy, and adjuvant chemotherapy [7,8], each of which can compromise brain integrity and function. Recent advancements in treatment have significantly improved survival rates, making the long-term consequences of cancer and cancer treatment on cognitive functioning an increasing concern [9,10].

Medulloblastoma survivors often exhibit a distinct neuropsychological profile characterized by deficits in processing speed, working memory, and executive functions. Other commonly affected domains include attention and visuospatial skills, which are critical for academic achievement, particularly in reading and mathematics [11]. To assess these cognitive alterations, survivors are often given a battery of neuropsychological tests. The Wechsler Intelligence Scale for Children (WISC) is frequently used to measure overall cognitive functioning [12], while other norm-referenced measures, such as the NEPSY-II battery, are applied to evaluate specific neuropsychological domains, such as attention, executive function, language, memory, and visuospatial processing [13].

The cognitive impairment experienced by medulloblastoma survivors arises from a complex interplay of factors. Both radiation therapy and chemotherapy, essential components of medulloblastoma treatment, induce significant structural and functional changes in the brain. These alterations, including white matter necrosis, disruptions in neurogenesis, vascular abnormalities, demyelination, neuroinflammation, and disturbances in neuronal connectivity, contribute to observed cognitive deficits [14]. The intricate relationship between treatment factors (such as dose, timing, and modality) and individual patient characteristics necessitates a personalized approach to cognitive rehabilitation. Late radiation therapy-induced changes in the brain can contribute to cognitive decline six months or later after treatment initiation [15]. This decline persists over time, potentially exacerbating and imposing significant challenges to survivors’ well-being and functional independence [15,16]. The trajectory of cognitive deficits in medulloblastoma survivors is concerning, as late-onset impairments can progressively worsen over the years, with a widening in the gap in skills between survivors and their same-aged peers [16].

Given the profound impact of cognitive impairment on the long-term outcomes and quality of life of medulloblastoma survivors, there is an urgent need for comprehensive assessment and intervention strategies. Early identification of cognitive deficits, coupled with targeted interventions, can mitigate the adverse effects and enhance functional outcomes [17]. Moreover, ongoing monitoring and longitudinal follow-up are essential to detect late-emerging cognitive changes and provide timely interventions to optimize survivors’ cognitive health and well-being.

Medulloblastoma survivors often experience significant cognitive impairments and reading challenges due to the tumor and its treatment. Studies have shown that these cognitive deficits can persist long-term, affecting the survivors’ quality of life and academic performance. For instance, children treated for medulloblastoma exhibit marked difficulties in processing speed and working memory, which are critical for reading [11,18]. Processing speed is estimated to be the most affected domain five years post-diagnosis [11]. Factors influencing cognitive outcomes include age at diagnosis, treatment intensity, and posterior fossa syndrome [11,19]. Such findings underscore the importance of early intervention and continuous support for these individuals [18,19].

Reading is a crucial cognitive skill that can be adversely affected by medulloblastoma and its treatment. Survivors often encounter reading difficulties, which can significantly impact long-term cognitive and psychological well-being, including academic performance and independent daily functioning. As these challenges persist and accumulate, survivors may increasingly face social isolation and rely more heavily on their network of caregivers [17]. This decline highlights the urgent need for effective cognitive interventions tailored to this population, including interventions focused on improving reading abilities. To address this, it is vital to understand the neurofunctional changes associated with reading impairment in medulloblastoma patients and how these changes evolve over time. Such findings are crucial for the development of targeted interventions and support services aimed at mitigating adverse effects and optimizing long-term outcomes for survivors.

Reading is a complex ability that involves the interplay of skills across multiple stages of information processing, encompassing the identification of visual patterns from retinotopic representation (orthography) and their conversion into spoken sounds (phonology) and meaning (semantics) within the brain [20]. Proficient readers demonstrate the ability to directly access the semantic meaning of words during reading, gradually minimizing reliance on slower mechanisms like phonological mediation as reading skills develop. The translation of visual symbols into sounds and meanings relies on a network of brain areas, with the ventral occipitotemporal cortex playing a crucial role [21,22]. The ventral occipitotemporal cortex, also known as the ventral visual pathway, performs visual object recognition, encompassing letters and words, where neural representations progress from retinotopic object features to categorical and individual objects [23]. Within the ventral visual pathway, the fusiform gyrus is involved in recognizing letters and words [24,25] with left-lateralization attributed to the distinct interconnections between the visual system and left-lateralized temporal language areas [25].

The “double deficit” hypothesis, as described by Wolf and Bowers [26], proposes that reading difficulties can arise from two distinct but interconnected factors: deficits in phonological awareness and deficits in rapid automatized naming (RAN). Children who exhibit deficits in both phonological awareness and RAN tend to experience more severe reading impairments compared to children with deficits in only one of these areas or none [27]. RAN measures retrieval automaticity, which correlates with reading fluency, and involves rapidly naming stimuli, such as letters, numbers, or colors, typically organized in a structured array. This enables researchers to assess the speed and efficiency of cognitive processing as well as the integrity of neural networks involved in language and visual processing [28].

During a RAN task, brain activation includes both left and right occipital-temporal regions for any of the three following conditions: color, number, and letter [8]. By combining behavioral measures of rapid naming with neuroimaging techniques, it is possible to elucidate the neural basis of cognitive impairments in particular conditions, such as medulloblastoma survivors [16], and develop targeted interventions to mitigate their impact on survivors’ cognitive health and overall well-being.

In this longitudinal study, we investigate neurofunctional changes related to reading in medulloblastoma survivors compared to healthy controls using functional magnetic resonance imaging (fMRI) during RAN tasks over three annual visits. Specifically, we aim to determine whether the reading difficulties observed in medulloblastoma survivors are due to disruptions in the ventral visual system of the brain. Our hypotheses are that (1) medulloblastoma survivors will show significant differences in ventral visual system activation during orthographic processing tasks compared to controls and that (2) these differences will correlate with the severity of cognitive impairments in reading-related tasks. To test these hypotheses, we assess the function of the ventral visual system in both groups using statistical and machine learning techniques, including the Fisher score for feature selection and support vector machines for classification [29,30].

## 2. Methods

### 2.1. Participants

This fMRI study (FNIMB1) conducted at St. Jude Children’s Research Hospital received approval from the institutional review board (IRB). Before participation, each participant gave informed assent, and written consent was obtained from the participant’s guardian if the participant was under 18 or from the participant if the participant was 18 or older. The study design incorporated a series of three fMRI sessions alongside reading assessments conducted at yearly intervals. During the initial fMRI session, medulloblastoma survivors were, on average, 2.5 years post-diagnosis, with a range from 2 to 4 years.

On the first visit (TP1), the study included 50 (18 females; 32 males) medulloblastoma survivors (mean = 13.5 years; SD = 3.9) and 50 (18 females; 32 males) control participants (mean = 13.5 years; SD = 4.8). Subsequently, 36 (14 females; 22 males) of the medulloblastoma survivors (mean = 14.4 years; SD = 4.3) and 36 (14 females; 22 males) of the controls (mean = 14.4 years; SD = 4.9) participated in the second visit (TP2), while 21 (7 females; 14 males) of the medulloblastoma survivors (mean = 16.1 years; SD = 3.5) and 21 (7 females; 14 males) of the controls (mean = 16.1 years; SD = 4.9) joined the third visit (TP3). Variations in participation numbers across visits were influenced by factors such as reluctance to engage in fMRI procedures, failure to adhere to the fMRI study protocol, challenges in task completion, and disease progression. For all cases, the group of participants in subsequent visits was a subset of the previous visit’s group.

All healthy children enrolled in the study were exclusively native English speakers and had no prior history of central nervous system injury or disease, attention deficits, or learning disabilities. Additionally, they did not present with any significant physical, neurological, or psychiatric conditions. They had not undergone treatment with psychostimulants or psychotropic medications within the two weeks preceding MRI or neuropsychological assessments, and they were not utilizing orthodontic appliances during the evaluation period.

### 2.2. Reading Scores

Reading abilities were assessed using the Woodcock-Johnson III Diagnostic Reading Battery [31], a comprehensive tool accounting for different aspects of reading proficiency, that includes letter-word identification, reading fluency, passage comprehension, word attack, spelling of sounds, sound awareness, and reading vocabulary (Table 1). Reading scores were standardized by age before being used in statistical analysis and pattern classification.

### 2.3. Functional Magnetic Resonance Imaging (fMRI)

Prior to the fMRI exam, structural MRI scans were conducted for all participants to ensure there were no gross anatomical abnormalities. Specifically, all medulloblastoma survivors in the study had undergone partial cerebellum resection, which is not considered an abnormality within this group. These scans were used only during the coregistration and normalization processes of the fMRI data to enhance the accuracy of our subsequent analyses, as detailed later in the manuscript.

fMRI data were acquired using a Siemens Trio 3T scanner (Siemens Medical Solutions, Erlangen, Germany), employing the following parameters: a single-shot T2*-weighted EPI sequence, repetition time (TR) of 2 s, echo time (TE) of 30 ms, field of view (FOV) of 192 mm, matrix size of 64 × 64, 32 slices with a slice thickness of 3.5 mm, and bandwidth of 2055 Hz/pixel.

The RAN task was administered using a standardized test protocol commonly utilized to assess cognitive processes involved in fluent reading [32]. During each task condition, participants were presented with a screen displaying five lines, each containing ten color squares, numbers, or letters. Participants were instructed to silently and rapidly name each line, press a button after naming each line, and start over if they reached the end before the stimulus screen changed. Stimulus blocks were presented for 20 s each and repeated three times in the pattern of NULL—Color—Number—Letter (Figure 1). A total of 135 image volumes were acquired for the RAN task. During the NULL condition, participants were instructed to focus on an array of plus signs (‘+’) to maintain visual consistency. Stimulus delivery and timing of both stimuli and responses were controlled using Presentation^®^ software Version 16.2 (https://www.neurobs.com/, accessed on 3 September 2024). Visual stimuli were presented on a rear-mounted screen and viewed through a mirror fixed to the head coil. The “Letter > Color” was selected as the primary contrast because it identifies brain regions involved in visual word recognition, reading, and language. This choice aligns with our goal of elucidating the specific neural mechanisms underlying cognitive impairments in medulloblastoma survivors.

Preprocessing of the fMRI images, including realignment, coregistration, normalization, and smoothing, was carried out using the SPM software Version 12 (http://www.fil.ion.ucl.ac.uk/spm/, accessed on 3 September 2024). For each visit, the voxel-wise BOLD difference “Letter > Color” was computed by subtracting the temporal-averaged BOLD signal for color stimuli from that of letter stimuli. This voxel-wise BOLD difference “Letter > Color” was then subjected to a feature selection procedure and utilized as multivariate features for support vector machine (SVM) classification between medulloblastoma survivors and controls. We chose to use SVMs due to their robustness in handling high-dimensional data and their effectiveness in binary classification tasks [29,30,33]. SVMs are particularly suitable for fMRI data analysis as they can manage the complex patterns of brain activity with a relatively small number of training samples. Previous studies [34,35,36,37] have successfully applied SVMs to classify neurocognitive patterns based on fMRI data, demonstrating their capability to distinguish different brain activity patterns.

A region-based feature selection process was used, wherein the voxel-wise BOLD difference “Letter > Color” was spatially averaged into region-wise BOLD difference “Letter > Color” based on the Neuromorphometrics atlas [38] (https://www.neuromorphometrics.com/, accessed on 3 September 2024). For each visit independently, the regional-wise BOLD difference “Letter > Color” of each subject underwent a feature selection filter utilizing the Fisher score [29,30] to estimate signal differences between medulloblastoma survivors and control volunteers in each brain region. Brain regions that met or exceeded these thresholds at least once across the three visits were selected. At the outset, Fisher score thresholds ranging from 0.16 to 0.26 (maximum threshold with selected brain regions), with increments of 0.02, were considered. Subsequently, the optimal threshold (0.22) was determined based on the highest average SVM accuracy (across the 3 visits).

### 2.4. Leave-One-Out Cross-Validation

The voxel-wise BOLD difference “Letter > Color” features located within the selected brain regions were utilized for linear support vector machine (SVM) classification [39], with the parameter estimation C set to 1. SVM classifications were carried out using leave-one-out cross-validation [40]. Leave-one-out cross-validation is a specific form of k-fold, where the number of folds is equal to the number of subjects [29].

In the first analysis, SVM was employed for binary classification between medulloblastoma survivors and control volunteers, with each visit treated separately. Subsequently, within the medulloblastoma survivors’ group and including all 3 visits, SVM was utilized for binary classification between high performers (standardized score ≥ 100) and low performers (standardized score < 100) on each of the following seven neurocognitive tests independently: letter-word identification, reading fluency, passage comprehension, word attack, spelling of sounds, sound awareness, and reading vocabulary.

## 3. Results

To identify the optimal Fisher score threshold for selecting brain regions of interest from the Neuromorphometrics atlas [38], SVM classification was performed for binary classification between medulloblastoma survivors and control volunteers. The thresholds ranged from 0.16 to 0.26, with increments of 0.02. Among the tested thresholds, 0.22 demonstrated superior performance compared to others (Figure 2). Therefore, it was deemed as the optimal threshold for this study.

Four brain regions survived the threshold (0.22) at least once across the three visits, which occurred exclusively in TP3 for all of them. These regions, as defined by the Neuromorphometrics atlas [38] and their respective Fisher scores on the TP3, include the right superior occipital gyrus (Fisher score = 0.273), right middle occipital gyrus (Fisher score = 0.248), right inferior occipital gyrus (Fisher score = 0.242), and right occipital pole (Fisher score = 0.221). These regions are all situated within the right occipital lobe, forming a single cluster (Figure 2).

Subsequently, this threshold was utilized for further analysis. To further evaluate the BOLD pattern within this cluster, we spatially averaged the BOLD difference “Letter > Color” among all voxels inside the cluster and grouped them by either medulloblastoma survivors or controls, for each visit separately. As a result, we observed a significant difference (*p*-value = 1.8 × 10^−3^) between medulloblastoma survivors and controls, exclusively in TP3 (Figure 3).

As this cluster is completely located within the right occipital lobe, we decided to investigate the BOLD patterns within the homologous contralateral region in the left occipital lobe, which is the cluster containing the left superior occipital gyrus, left middle occipital gyrus, left inferior occipital gyrus, and left occipital pole. As a result, we observed again a significant difference (*p*-value = 1.9 × 10^−2^) between medulloblastoma survivors and controls exclusively in TP3 (Figure 3).

SVM classification between medulloblastoma survivors and control volunteers based on the BOLD activity in the pre-selected brain regions (right superior occipital gyrus, right middle occipital gyrus, right inferior occipital gyrus, and right occipital pole) in the right occipital lobe achieved an accuracy of 57% in TP1, 75% in TP2, and 79% in TP3 (Table 2).

The weight vector maps from the SVM model reveal sparse voxel contributions during TP1 and TP2. Conversely, TP3 exhibits a pronounced concentration of significant voxel contributions, particularly notable between the inferior and middle occipital gyrus, lingual gyrus, and fusiform gyrus, all within the right hemisphere (Figure 4). For TP3, only one cluster, containing 190 voxels, survived the threshold of a minimum volume of 50 voxels, with each voxel weight vector having an absolute value of 0.02 or greater.

To explore the neurocognitive effects of identified BOLD alterations within the medulloblastoma survivors’ cohort, we utilized SVM analysis for binary classification, stratifying individuals into high performers (standardized score ≥ 100) and low performers (standardized score < 100) across seven distinct neurocognitive tests: letter-word identification, reading fluency, passage comprehension, word attack, spelling of sounds, sound awareness, and reading vocabulary. Our SVM model demonstrated above-chance accuracy in discerning between high and low performers, specifically in the sound awareness, reading fluency, and word attack tests (Table 3).

## 4. Discussion

Understanding the neurofunctional alterations associated with medulloblastoma and its treatment is crucial for optimizing clinical management and improving outcomes for survivors. In this study, we investigated the neurofunctional differences between medulloblastoma survivors and healthy controls using SVM classification for fMRI data analysis. Our findings shed light on the temporal dynamics of brain alterations and their implications for cognitive functioning in this population.

The observed improvement in SVM classification accuracy, from 57% in TP1 to 79% in TP3, reflects a progressive divergence in brain activity patterns related to orthographic processing between medulloblastoma survivors and controls over time. The observation that these neurofunctional alterations are noticed primarily at the long-term (TP3) aligns with previous studies demonstrating delayed effects of cancer treatment on brain structure and function [14,15,43,44,45,46]. The lower accuracy in TP1 may be attributed to the more subtle and less pronounced differences in brain activity shortly after treatment, when neurofunctional changes are still emerging and have not fully manifested. These early stage changes are harder to detect, leading to reduced classification accuracy.

As time progresses, the cumulative effects of cancer treatment, particularly radiation therapy, become more pronounced, contributing to the increased accuracy observed at TP2 and TP3. Radiation therapy, a common treatment for medulloblastoma, can induce progressive alterations in brain tissue, including changes in white matter integrity and cortical thickness, which often develop gradually and may take years to fully manifest. Moreover, cognitive deficits in medulloblastoma survivors frequently emerge or worsen over time due to ongoing neuroinflammatory processes and neuroplasticity changes that continue long after treatment ends [47,48]. This gradual evolution of brain structure and function aligns with the improvement in classification accuracy seen at TP2 and TP3, when neurofunctional differences between survivors and controls are more pronounced.

The significant differences observed in brain regions such as the right superior occipital gyrus, right middle occipital gyrus, right inferior occipital gyrus, and right occipital pole during the rapid naming test align with previous literature highlighting the involvement of the bilateral occipital regions in visual processing and the ventral visual pathway in letter recognition [9,49]. Their contralateral counterparts play a pivotal role in orthographic processing, which involves recognizing and interpreting visual symbols such as letters [50]. Consequently, our findings suggest a reduced involvement of the right ventral visual pathway in the rapid naming of letters task. This discovery prompts us to speculate about the potential inhibition of contralateral brain regions during rapid orthographic processing, possibly as an effort to uphold the left-lateralization of orthographic processing, notwithstanding potential disruptions in bilateral neuropathways associated with normal orthographic processing.

The SVM weight vector maps, also known as effect maps [51], provide valuable insights into the neurofunctional differences between medulloblastoma survivors and controls. The observed patterns of voxel contributions highlight the involvement of specific brain regions, including the inferior and middle occipital gyrus, lingual gyrus, and fusiform gyrus, in mediating cognitive processes related to reading and visual processing. For instance, during the rapid naming test, individuals are required to quickly name visual stimuli such as colors or letters presented on a screen. The inferior and middle occipital gyrus contribute to the rapid identification and naming of these visual stimuli, as they are involved in processing visual information and recognizing familiar objects and symbols [21,22]. Additionally, the lingual gyrus and fusiform gyrus, located in the occipital lobe, are implicated in various aspects of visual processing, including word recognition and reading [24,25]. They may play roles in processing visual stimuli and converting them into meaningful representations for naming, particularly in tasks requiring the naming of written words or letters [23].

The neurocognitive effects of the identified BOLD alterations were further elucidated using SVM analysis of high and low performers on neurocognitive tests. Our SVM model demonstrated above-chance accuracy in distinguishing between high and low performers among medulloblastoma survivors, particularly in the sound awareness, reading fluency, and word attack tests. This highlights the potential utility of neurofunctional biomarkers in predicting cognitive outcomes and tailoring interventions for medulloblastoma survivors. Expanding upon the predictive ability and interventions could amplify the potential clinical impact of these findings. For instance, it could inform decisions regarding sparing certain brain regions from radiotherapy or recommend specific types of reading interventions.

The integrity of the right occipital cortex, including regions such as the lingual gyrus and fusiform gyrus, may affect sound awareness, reading fluency, and word attack performance through their connections linking visual perception areas in the occipital lobe with auditory processing regions in the temporal lobe. The lingual gyrus and fusiform gyrus, located in the medial occipitotemporal region spanning both lobes, are positioned adjacent to the primary visual cortex. They play a crucial role in facilitating interactions between visual and auditory stimuli through multisensory integration and cross-modal processing [24]. Alterations in these regions among medulloblastoma survivors may disrupt their ability to integrate auditory and visual information, thereby impacting sound awareness, reading fluency, and word attack performance.

Overall, our findings underscore the complex interplay between cancer treatment, neurofunctional alterations, and cognitive outcomes in medulloblastoma survivors. Future research should investigate the underlying mechanisms driving these neurofunctional changes and explore targeted interventions to mitigate cognitive impairments in this population.

## 5. Conclusions

In conclusion, our study sheds light on the temporal evolution of BOLD signal changes in medulloblastoma survivors and their implications for cognitive functioning. Through a comprehensive investigation utilizing fMRI and neurocognitive assessments, we investigated the complex interplay among cancer treatment, neurofunctional alterations, and cognitive outcomes in medulloblastoma survivors.

The progressive divergence in brain activity patterns between medulloblastoma survivors and healthy controls over time emphasizes the dynamic nature of post-treatment brain function and the delayed effects of cancer treatment on brain structure and function. Importantly, our findings highlight the potential utility of neurofunctional biomarkers, such as those identified through SVM classification of fMRI data, in predicting cognitive outcomes and tailoring interventions for medulloblastoma survivors.

Specifically, the observed alterations in brain regions involved in visual processing and orthographic recognition during rapid naming tasks suggest potential disruptions in the ventral visual pathway, which is associated with normal orthographic processing. These findings prompt further investigations into the underlying mechanisms driving these neurofunctional changes and their impact on cognitive processes such as reading and sound awareness.

Moreover, our study underscores the importance of the early identification of cognitive deficits and the need for tailored interventions to mitigate their impact on survivors’ cognitive health and overall well-being. By understanding the neurofunctional basis of cognitive impairments in medulloblastoma survivors, clinicians and researchers can develop targeted interventions to optimize cognitive outcomes and improve the quality of life for this population.

Our sample size, although representative, may limit the generalizability of our findings to broader populations of medulloblastoma survivors. We acknowledge potential confounding factors, such as socioeconomic status and pre-treatment cognitive functioning, were not accounted for in our analysis. Furthermore, while our sample is diverse, it may not capture the full range of demographic and clinical characteristics observed in the wider medulloblastoma population. The design of the fMRI RAN task could also be improved with a sufficient duration for the rest condition between the letter, color, and number contrasts to allow the BOLD activity to return to baseline before the start of the next condition. These factors should be considered when interpreting the results and generalizing the findings.

Future studies may focus on elucidating the underlying mechanisms driving neurofunctional alterations in medulloblastoma survivors, exploring the efficacy of targeted interventions in mitigating cognitive impairments, and investigating the long-term neurocognitive trajectories of survivors across different stages of post-treatment life. By addressing these research priorities, we can advance our understanding of the neurobiological basis of cognitive impairments in medulloblastoma survivors and develop effective strategies to optimize their cognitive health and overall well-being.

## Figures and Tables

**Figure 1 brainsci-14-00904-f001:**
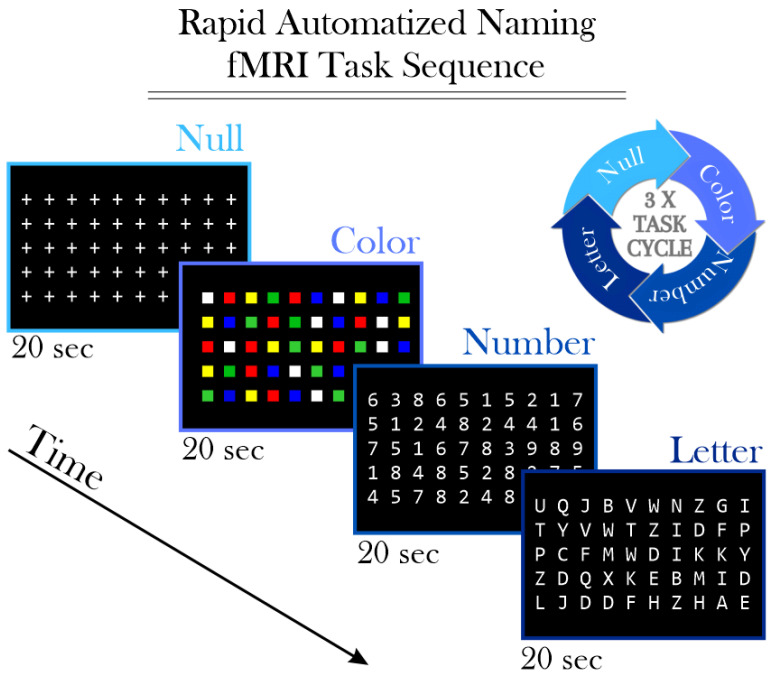
This diagram illustrates the sequence of tasks in the fMRI experiment, represented as a cyclic process. Null: Participants focus on plus signs (‘+’) for 20 s; Color: Participants silently name five lines of color squares for 20 s; Number: Participants silently name five lines of numbers for 20 s; Letter: Participants silently name five lines of letters for 20 s. This sequence is repeated three times. The first cycle starts 10 s after the scanning process begins, while the last cycle finishes 20 s before the scanning ends. This results in a total task duration of 270 s.

**Figure 2 brainsci-14-00904-f002:**
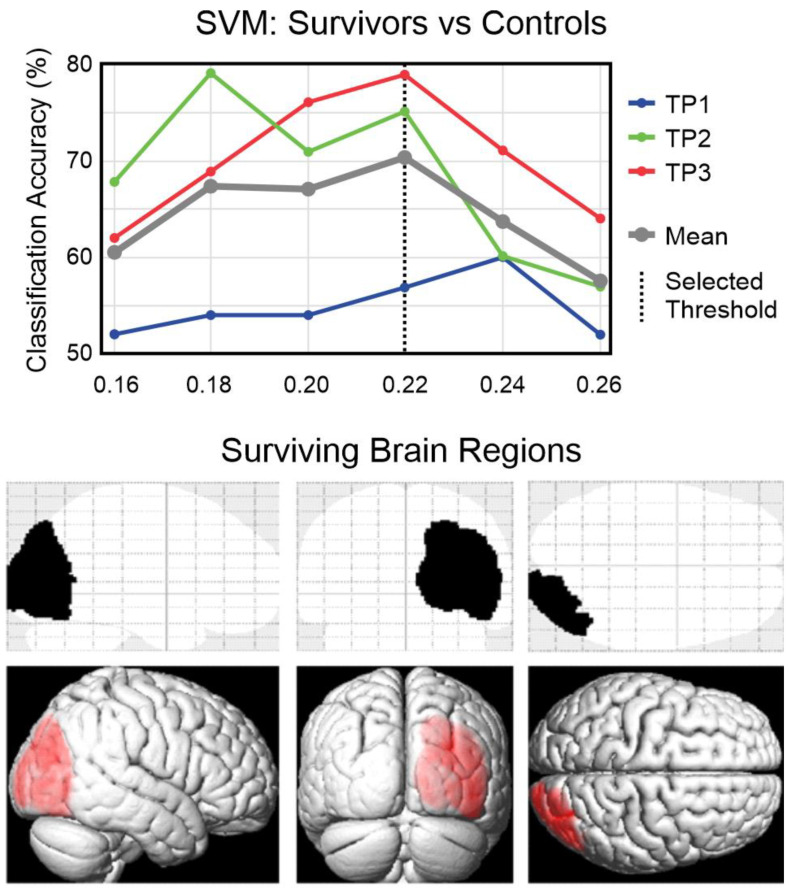
SVM accuracy of binary classification (medulloblastoma survivors vs. control volunteers), assessed separately for each visit (TP1, TP2, and TP3), along with their average accuracy. The classification was conducted for various Fisher score thresholds, ranging from 0.16 to 0.26 with increments of 0.02. A single cluster (glass image: black; 3D rendering view: red) formed by the four brain regions survived the optimal threshold (0.22) at least once across the three visits. These regions are the right superior occipital gyrus, right middle occipital gyrus, right inferior occipital gyrus, and right occipital pole (all situated within the right occipital lobe, as defined by the Neuromorphometrics atlas [38]). Top: Glass brain images. Vertical Center: Glass images. Bottom: 3D rendering view; Left: Left sagittal view. Horizontal Center: Posterior coronal view. Right: Superior axial view.

**Figure 3 brainsci-14-00904-f003:**
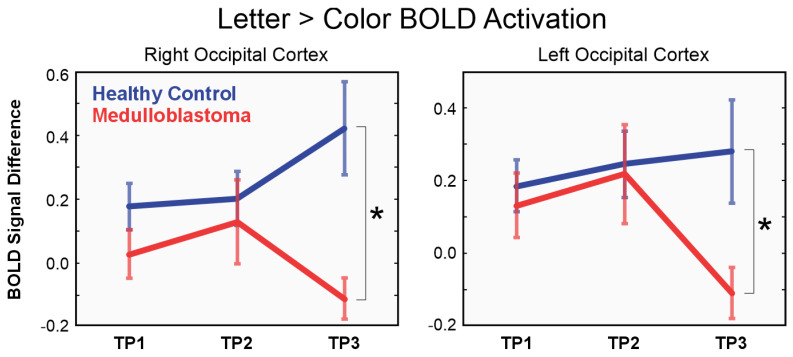
BOLD difference “Letter > Color” cluster averaged and grouped by participant group (medulloblastoma survivors depicted in red; controls in blue) for visits TP1, TP2, and TP3. The stars * indicate significant differences (right occipital cortex: *p* = 1.8 × 10^−3^; left occipital cortex: *p* = 1.9 × 10^−2^) between medulloblastoma survivors and controls, observed exclusively in TP3. (**Left Graph**): Single cluster on right occipital cortex that survived the optimal threshold. (**Right Graph**): Contralateral homolog cluster on the left occipital cortex.

**Figure 4 brainsci-14-00904-f004:**
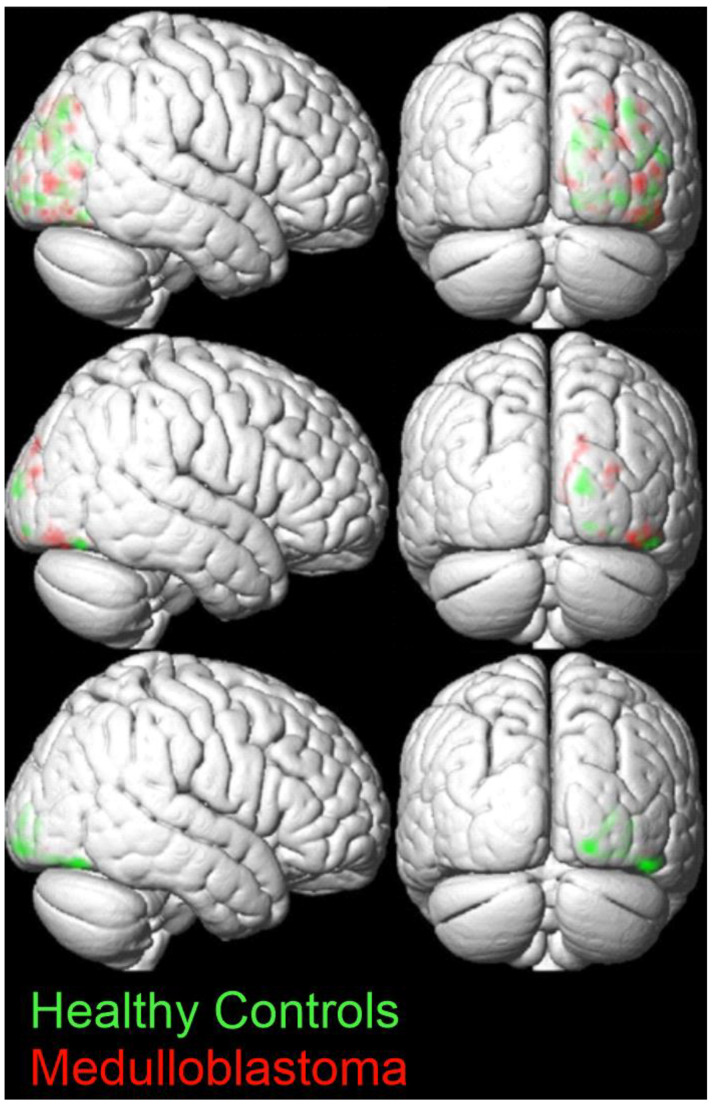
Right sagittal views (**left**) and posterior coronal views (**right**) of the weight vector maps for visits TP1 (**top**), TP2 (**middle**), and TP3 (**bottom**). Green indicates voxel-wise BOLD difference “Letter > Color” weight vectors positively associated with the control group, while red indicates weight vectors positively associated with the medulloblastoma survivors’ group. The clusters are thresholded based on a minimum volume of 50 voxels, with each voxel weight vector required to have an absolute value of 0.02 or greater.

**Table 1 brainsci-14-00904-t001:** Description of the acquired reading scores.

Task	Description
Letter-Word Identification	Assesses the ability to identify and read isolated letters and words accurately.
Reading Fluency	Measures the speed and accuracy of reading simple sentences within a limited time frame.
Passage Comprehension	Evaluates the ability to comprehend the meaning of written passages by filling in missing words.
Word Attack	Tests the skill of decoding and pronouncing unfamiliar pseudo-words, focusing on phonetic decoding abilities.
Spelling of Sounds	Assesses the ability to spell words by dictation, focusing on phonetic spelling skills.
Sound Awareness	Measures phonological awareness, including tasks such as rhyming, segmenting, and manipulating sounds.
Reading Vocabulary	Evaluates the understanding and knowledge of word meanings through synonyms, antonyms, and analogies.

**Table 2 brainsci-14-00904-t002:** SVM classification results for distinguishing between medulloblastoma survivors (positive label) and control volunteers (negative label), including accuracy, sensitivity, and specificity. The Monte Carlo *p*-value [41] indicates the likelihood of achieving equal or higher accuracy with shuffled label data after 1000 shuffles. An asterisk (*) marks accuracies with a *p*-value below 0.05.

Visit	Accuracy	Sensitivity	Specificity	*p*-Value
TP1	57%	60%	54%	118/1000
TP2	75%	69%	81%	1/1000 (*)
TP3	79%	81%	76%	3/1000 (*)

**Table 3 brainsci-14-00904-t003:** SVM classification results (accuracy, sensitivity, specificity, and balanced accuracy) for distinguishing between high performing (standardized score ≥ 100) and low performing (standardized score < 100) medulloblastoma patient visits on each of the following seven neurocognitive tests independently: letter-word identification, reading fluency, passage comprehension, word attack, spelling of sounds, sound awareness, and reading vocabulary. Size (H/L) indicates the dataset size for high performing (H) and low performing (L). Balanced accuracy is calculated as the arithmetic mean of sensitivity and specificity [42]. The Monte Carlo *p*-value [41] indicates the likelihood of achieving equal or higher accuracy with shuffled label data after 1000 shuffles. An asterisk (*) denotes accuracies with a *p*-value below 0.05.

Neurocognitive Test	Size (H/L)	Accuracy	Sensitivity	Specificity	Balanced Accuracy	*p*-Value
Reading Fluency	28/61	68.5%	42.9%	80.3%	61.6%	12/1000 (*)
Passage Comprehension	45/47	50.0%	44.4%	55.3%	49.9%	788/1000
Word Attack	51/41	65.2%	64.7%	65.9%	65.3%	16/1000 (*)
Spelling of Sounds	48/44	51.1%	50.0%	52.3%	51.2%	719/1000
Sound Awareness	51/41	67.4%	70.6%	63.4%	67.0%	4/1000 (*)
Reading Vocabulary	53/39	55.4%	64.2%	43.6%	53.9%	437/1000
Letter-Word Identification	50/42	51.1%	52.0%	50.0%	51.0%	719/1000

## Data Availability

The data presented in this study are available on request from the corresponding author. The data are not publicly available due to privacy.

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
