# Peer review of "Functional MRI Assessment of Brain Activity Patterns Associated with Reading in Medulloblastoma Survivors"

_brainsci, 2024, doi:10.3390/brainsci14090904_

Round 1

Reviewer 1 Report

Comments and Suggestions for Authors

The paper titled "Functional MRI Assessment of Brain Activity Patterns Associated with Reading in Medulloblastoma Survivors" investigates the cognitive impacts of medulloblastoma treatment on children, focusing on reading impairments. Utilizing functional MRI (fMRI) during rapid automatized naming (RAN) tasks, the study observes longitudinal changes in brain activity among survivors compared to healthy controls over three annual visits. The results highlight progressive neurofunctional divergence, particularly in regions involved in visual processing and orthographic recognition, such as the occipital gyrus. These findings suggest delayed treatment effects on brain function and emphasize the need for early identification and intervention to address cognitive deficits in survivors, thereby improving their long-term cognitive health and quality of life.

Here below are some comments:

  • Clarify Figures: Enhance the clarity of figures and ensure all labels and legends are easily readable.
  • Expand Literature Review: Include more recent studies to provide a broader context for the findings.
  • .
  • Limitations: While the study's limitations are acknowledged, a more detailed discussion on potential confounding factors and the generalizability of the findings would be beneficial.

Author Response

Comments 1: The paper titled "Functional MRI Assessment of Brain Activity Patterns Associated with Reading in Medulloblastoma Survivors" investigates the cognitive impacts of medulloblastoma treatment on children, focusing on reading impairments. Utilizing functional MRI (fMRI) during rapid automatized naming (RAN) tasks, the study observes longitudinal changes in brain activity among survivors compared to healthy controls over three annual visits. The results highlight progressive neurofunctional divergence, particularly in regions involved in visual processing and orthographic recognition, such as the occipital gyrus. These findings suggest delayed treatment effects on brain function and emphasize the need for early identification and intervention to address cognitive deficits in survivors, thereby improving their long-term cognitive health and quality of life.

Here below are some comments:

Clarify Figures: Enhance the clarity of figures and ensure all labels and legends are easily readable.

Response 1: Thank you for your advice. We have revised all the figures to enhance their readability and ensure that all labels and legends are now more clearly visible.

Comments 2:  Expand Literature Review: Include more recent studies to provide a broader context for the findings.

Response 2: Thank you for your suggestion. In response, we have expanded the introduction sections to incorporate additional references that provide a broader context for our findings. For instance, we included the following:

Medulloblastoma is the most common malignant brain tumor in children, accounting for approximately 20% of all pediatric brain tumors [1]. It predominantly affects the pediatric population, with a 10-fold higher frequency than in adults [2], particularly in children between the ages of 2 and 8 years [3]. Medulloblastoma primarily arises in the cerebellum or posterior brainstem, located within the posterior fossa [4]. The cerebellum is known to interact with supratentorial somatomotor circuits to coordinate movement, and increasing evidence suggests it also plays a role in coordinating cognitive processes with the cerebrum [5,6]. Clinically, medulloblastoma often presents with symptoms related to increased intracranial pressure, such as headaches, nausea, vomiting, and balance difficulties, due to its location in the posterior fossa [1]. In younger children, irritability and developmental delay may also be observed. Treatment for medulloblastoma includes maximal surgical resection, radiation therapy, and adjuvant chemotherapy [7,8], each of which can compromise brain integrity and function. Recent advancements in treatment have significantly improved survival rates, making the long-term consequences of cancer and cancer treatment on cognitive functioning an increasing concern [9,10].

Medulloblastoma survivors often exhibit a distinct neuropsychological profile characterized by deficits in processing speed, working memory, and executive functions. Other commonly affected domains include attention and visuospatial skills, which are critical for academic achievement, particularly in reading and mathematics [11]. To assess these cognitive alterations, survivors are often given a battery of neuropsychological tests. The Wechsler Intelligence Scale for Children (WISC) is frequently used to measure overall cognitive functioning [12], while other norm-referenced measures such as the NEPSY-II battery are applied to evaluate specific neuropsychological domains such as attention, executive function, language, memory, and visuospatial processing [13].”

Additionally, we added this:

“Medulloblastoma survivors often experience significant cognitive impairments and reading challenges due to the tumor and its treatment. Studies have shown that these cognitive deficits can persist long-term, affecting the survivors' quality of life and academic performance. For instance, children treated for medulloblastoma exhibit marked difficulties in processing speed and working memory, which are critical for reading [11, 18]. Processing speed is estimated to be the most affected domain five years post-diagnosis [11]. Factors influencing cognitive outcomes include age at diagnosis, treatment intensity, and posterior fossa syndrome [11, 19]. Such findings underscore the importance of early intervention and continuous support for these individuals [18,19].”

Comment 3:  Limitations: While the study's limitations are acknowledged, a more detailed discussion on potential confounding factors and the generalizability of the findings would be beneficial.

Response 3: Thank you for your suggestion. We have expanded the conclusion to include a detailed discussion of the study's limitations. The revised conclusion now addresses the following points:

“Our sample size, although representative, may limit the generalizability of our findings to broader populations of medulloblastoma survivors. We acknowledge potential confounding factors, such as socioeconomic status and pre-treatment cognitive functioning, were not accounted for in our analysis. Furthermore, while our sample is diverse, it may not capture the full range of demographic and clinical characteristics observed in the wider medulloblastoma population. The design of the fMRI RAN task could also be improved with sufficient duration of the rest condition between the letter, color and number contrasts to allow for the return of the BOLD activity to the baseline before the start of the next condition. These factors should be considered when interpreting the results and generalizing the findings.”

Reviewer 2 Report

Comments and Suggestions for Authors

In this study, the authors handle the medulloblastoma survivors and search the brain activity patterns. The study is quite important, and the studies on this topic are worth to support.  However, there are some concerns about the paper, as given below:

·         Please give more details and refer other related studies on page 2, where you mention the cognitive impairment and reading challenges with blastoma (the paragraphs given with reference [8].)

·         The authors should give the motivation for using SVM in detail.

·         Please give the other missing information about the participants, i.e., on the first visit 50 participants were included, where 18 of them are female with 13..5 y.o and 3.9 std. dev. What about the males? Like that, give the details for the second and the third visits. Another question is, does the control group on first, second and third visits consist of the same participants or they changed?

·      Did you perform any structural mri scan before the task?

·         It would be better to give the fMRI task as a flow chart to improve the understandability of the paper and the task.

·         It would be much better to see the other contrasts apart from the letter>color.  In the analysis, we cannot see the effect of the “number, so here a question appears that what is the contribution or the effect of the number session in the task?

·         It is not clear in Figure 1 whether they belong to the survivors or controls or both. It would be better to see each group separately on the same graphic, or two graphics next to each other for the survivors and controls.

·         please provide the glass brain images for the four brain regions on p.6.

·         The accuracy analysis for the neurocognitive tests should be supported for each visit. (Letter-Word Identification, Reading Fluency, Passage Comprehension, Word Attack, Spelling of Sounds, Sound Awareness, Reading Vocabulary)

Author Response

Comments 1: In this study, the authors handle the medulloblastoma survivors and search the brain activity patterns. The study is quite important, and the studies on this topic are worth to support.  However, there are some concerns about the paper, as given below:

Please give more details and refer other related studies on page 2, where you mention the cognitive impairment and reading challenges with blastoma (the paragraphs given with reference [8].)

Response 1: Thank you for this advice. We have expanded the section on cognitive impairment and reading challenges in medulloblastoma survivors. Additional details and references to related studies have been included to provide a more comprehensive background. Specifically, we have added the following text:

“Medulloblastoma survivors often experience significant cognitive impairments and reading challenges due to the tumor and its treatment. Studies have shown that these cognitive deficits can persist long-term, affecting the survivors' quality of life and academic performance. For instance, children treated for medulloblastoma exhibit marked difficulties in processing speed and working memory, which are critical for reading [11, 18]. Processing speed is estimated to be the most affected domain five years post-diagnosis [11]. Factors influencing cognitive outcomes include age at diagnosis, treatment intensity, and posterior fossa syndrome [11, 19]. Such findings underscore the importance of early intervention and continuous support for these individuals [18,19].”

Comments 2:  The authors should give the motivation for using SVM in detail.

Response 2: Thank you for this suggestion. We have provided a detailed rationale for using Support Vector Machines (SVM) in our study:

We chose to use SVM due to their robustness in handling high-dimensional data and their effectiveness in binary classification tasks [29, 30, 33]. SVMs are particularly suitable for fMRI data analysis as they can manage the complex patterns of brain activity with a relatively small number of training samples. Previous studies [34-37] have successfully applied SVMs to classify neurocognitive patterns based on fMRI data, demonstrating their capability to distinguish between different brain activity patterns.”

Comments 3:  Please give the other missing information about the participants, i.e., on the first visit 50 participants were included, where 18 of them are female with 13..5 y.o and 3.9 std. dev. What about the males? Like that, give the details for the second and the third visits. Another question is, does the control group on first, second and third visits consist of the same participants or they changed?

Response 3: We appreciate your suggestion to provide additional details regarding the participants. Participants were 18 females and 32 males in the first visit. Although we initially believed that the number of males could be inferred by subtracting the number of females (18) from the total participants (50), we have now explicitly included the number of males in the manuscript, as recommended. The mean age and standard deviation mentioned previously referred to the 50 total participants regardless of sex. We also have revised the manuscript to include gender distribution, age, and standard deviations for participants at each visit. Additionally, we clarified that the control group consisted of the same participants across all three visits. For all cases, the group of participants in subsequent visits was a subset of the previous visit's group. We have revised in the manuscript, accordingly:

On the first visit (TP1), the study included 50 (18 females; 32 males) medulloblastoma survivors (mean = 13.5 years; SD = 3.9) and 50 (18 females; 32 males) control participants (mean = 13.5 years; SD = 4.8). Subsequently, 36 (14 females; 22 males) of the medulloblastoma survivors (mean = 14.4 years; SD = 4.3) and 36 (14 females; 22 males) of the controls  (mean = 14.4 years; SD = 4.9) participated in the second visit (TP2), while 21 (7 females; 14 males) of the medulloblastoma survivors (mean = 16.1 years; SD = 3.5) and 21 (7 females; 14 males) of the controls (mean = 16.1 years; SD = 4.9) joined the third visit (TP3). Variations in participation numbers across visits were influenced by factors such as reluctance to engage in fMRI procedures, failure to adhere to the fMRI study protocol, challenges in task completion, and disease progression. For all cases, the group of participants in subsequent visits was a subset of the previous visit's group.”

Comments 4:  Did you perform any structural mri scan before the task?

Response 4: Thank you for this reminder. We have now specified in the manuscript that structural MRI scans were conducted for all participants before the task:

“Prior to the fMRI exam, structural MRI scans were conducted for all participants to ensure there were no gross anatomical abnormalities. Specifically, all medulloblastoma survivors in the study had undergone partial cerebellum resection, which is not considered an abnormality within this group. These scans were used only during the coregistration and normalization processes of the fMRI data to enhance the accuracy of our subsequent analyses, as detailed later in the manuscript.”

Comments 5:  It would be better to give the fMRI task as a flow chart to improve the understandability of the paper and the task.

Response 5: Thank you for this advice. A flow chart representing the fMRI task has been included in the manuscript (Figure 1).

Comments 6:  It would be much better to see the other contrasts apart from the letter>color.  In the analysis, we cannot see the effect of the “number, so here a question appears that what is the contribution or the effect of the number session in the task?

Response 6: The focus of this study is on the functional MRI assessment of brain activity patterns specifically associated with reading in medulloblastoma survivors. The "letter>color" contrast was chosen because it directly targets the neural processes underlying reading and literacy skills, which are of primary interest given the documented reading challenges in this population.

The "number" condition, while included in the task design for a broader cognitive assessment, did not show significant and consistent activations in the brain. Therefore, we have limited our analysis to the contrasts most relevant to understanding reading impairment and its associated brain activity patterns in medulloblastoma survivors.

Comments 7:  It is not clear in Figure 1 whether they belong to the survivors or controls or both. It would be better to see each group separately on the same graphic, or two graphics next to each other for the survivors and controls.

Response 7: The figure in question, now labeled as Figure 2, shows the SVM accuracy of binary classification (medulloblastoma survivors vs. control volunteers). We have updated both the figure and its legend to improve clarity and data interpretation.

Comments 8:  please provide the glass brain images for the four brain regions on p.6.

Response 8: Thank you for this advice. Glass brain images for the mentioned brain regions have been included in the manuscript (Figure 2). These images provide a clearer visualization of the brain regions of interest.

Comments 9: The accuracy analysis for the neurocognitive tests should be supported for each visit. (Letter-Word Identification, Reading Fluency, Passage Comprehension, Word Attack, Spelling of Sounds, Sound Awareness, Reading Vocabulary)

Response 9: We appreciate the suggestion to conduct accuracy analyses for each neurocognitive test across all three visits. However, splitting the data into three separate subgroups would substantially reduce the sample size for each visit. This reduction in sample size could significantly diminish statistical power, leading to results that might trend towards non-significance—not due to an absence of underlying neurocognitive effects, but simply due to the limitations imposed by the smaller sample size.

Moreover, applying multiple tests to these smaller subgroups would require a Bonferroni correction to control for Type I errors, further raising the significance threshold. This adjustment would increase the likelihood of failing to detect meaningful effects, even when they are present, as the thresholds for significance would become prohibitively high.

Given these constraints, we believe that our current approach, which aggregates the data across visits, strikes a more effective balance between statistical rigor and interpretability. By maintaining a sufficient sample size, we preserve the power needed to detect significant effects, ensuring that our findings more accurately reflect the neurocognitive consequences of neurofunctional changes in medulloblastoma survivors. This method allows us to provide a reliable and interpretable assessment, avoiding the risk of misinterpreting results due to statistical limitations rather than actual phenomena.

Reviewer 3 Report

Comments and Suggestions for Authors

Functional MRI Assessment of Brain Activity Patterns Associated with Reading in Medulloblastoma Survivors

I read the manuscript with interest and, according to me, the topic is innovative. You can find my appraisal, including concerns and suggestions, as follows.

Introduction: The introduction is well written. However, the definition of Medulloblastoma is quite vague and it needs to be defined in a better way. Moreover, information about the target population is needed. I suggest to add for the mean age of the patients and clarifying if adults can also be affected. According to a recent study, Medulloblastoma was found to affect children between 2 and 8 years. Similarly, the clinical manifestation is not clear in the introduction. I suppose that is mainly due to the fact that you are experts in this field.    However, this information is needed for readers who are not familiar with the above-mentioned disorder. You could take a look at https://doi.org/10.3389/fonc.2020.566822 (not mine and only a suggestion). Moreover, I suggest characterizing in a better way, the cognitive alterations observed in young patients and the neuropsychological profiles, including the tests used to assess the alterations in the patients (not only related to reading).  The hypotheses are clear.

Methods: The methods are clear and, the description is detailed appearing to be written according to COBIDAS or similar checklists. The longitudinal part of the design (mixed ) needs to be highlighted since you mentioned the “VISIT”. In this way, I suggest using a more common term like T1, T2, and T3. Maybe it is more intuitive, but it is an advice. The neuropsychological assessment is meticulous. The description of the task is simple, but it is not clear in which manner you controlled the time or repetition effect, which is crucial for fMRI studies. In general, SVM can limit this effect, but this info is needed. However, the method is innovative. I advise using an axial presentation instead of a 3D template if possible. Please, add or modify. Add, if possible the coordinates of the results.

The results are interesting, however, the accuracy, sensitivity, and specificity for VISIT 1 are lower than other VISITS.  

The Discussion is in line with the results, but I suggest discussing the results of the accuracy, sensitivity, and specificity for VISIT 1. However, I suggest improving the discussion in light of previous results on Medulloblastoma patients. For this reason, I also suggest improving the bibliographic part.

Moreover, I suggest adding the limitation of the study in the conclusion. 

Author Response

Comments 1: I read the manuscript with interest and, according to me, the topic is innovative. You can find my appraisal, including concerns and suggestions, as follows.

Introduction: The introduction is well written. However, the definition of Medulloblastoma is quite vague and it needs to be defined in a better way. Moreover, information about the target population is needed. I suggest to add for the mean age of the patients and clarifying if adults can also be affected. According to a recent study, Medulloblastoma was found to affect children between 2 and 8 years. Similarly, the clinical manifestation is not clear in the introduction. I suppose that is mainly due to the fact that you are experts in this field. However, this information is needed for readers who are not familiar with the above-mentioned disorder. You could take a look at https://doi.org/10.3389/fonc.2020.566822 (not mine and only a suggestion). Moreover, I suggest characterizing in a better way, the cognitive alterations observed in young patients and the neuropsychological profiles, including the tests used to assess the alterations in the patients (not only related to reading). The hypotheses are clear.

Response 1: Thank you for your comments. We have revised the introduction to provide a clearer definition of medulloblastoma.

We included in the manuscript this:

Medulloblastoma is the most common malignant brain tumor in children, accounting for approximately 20% of all pediatric brain tumors [1]. It predominantly affects the pediatric population, with a 10-fold higher frequency than in adults [2], particularly in children between the ages of 2 and 8 years [3]. Medulloblastoma primarily arises in the cerebellum or posterior brainstem, located within the posterior fossa [4]. The cerebellum is known to interact with supratentorial somatomotor circuits to coordinate movement, and increasing evidence suggests it also plays a role in coordinating cognitive processes with the cerebrum [5,6]. Clinically, medulloblastoma often presents with symptoms related to increased intracranial pressure, such as headaches, nausea, vomiting, and balance difficulties, due to its location in the posterior fossa [1]. In younger children, irritability and developmental delay may also be observed. Treatment for medulloblastoma includes maximal surgical resection, radiation therapy, and adjuvant chemotherapy [7,8], each of which can compromise brain integrity and function. Recent advancements in treatment have significantly improved survival rates, making the long-term consequences of cancer and cancer treatment on cognitive functioning an increasing concern [9,10].

Medulloblastoma survivors often exhibit a distinct neuropsychological profile characterized by deficits in processing speed, working memory, and executive functions. Other commonly affected domains include attention and visuospatial skills, which are critical for academic achievement, particularly in reading and mathematics [11]. To assess these cognitive alterations, survivors are often given a battery of neuropsychological tests. The Wechsler Intelligence Scale for Children (WISC) is frequently used to measure overall cognitive functioning [12], while other norm-referenced measures such as the NEPSY-II battery are applied to evaluate specific neuropsychological domains such as attention, executive function, language, memory, and visuospatial processing [13].”

Comments 2: Methods: The methods are clear and, the description is detailed appearing to be written according to COBIDAS or similar checklists. The longitudinal part of the design (mixed ) needs to be highlighted since you mentioned the “VISIT”. In this way, I suggest using a more common term like T1, T2, and T3. Maybe it is more intuitive, but it is an advice. The neuropsychological assessment is meticulous. The description of the task is simple, but it is not clear in which manner you controlled the time or repetition effect, which is crucial for fMRI studies. In general, SVM can limit this effect, but this info is needed. However, the method is innovative. I advise using an axial presentation instead of a 3D template if possible. Please, add or modify. Add, if possible the coordinates of the results.

Response 2: We appreciate your suggestion. We have now updated the manuscript accordingly, replacing "VISIT" with TP1, TP2, and TP3.

Regarding the control of time and repetition effects across the three annual measurements, our approach involved several steps. Initially, for the binary classification between medulloblastoma survivors and control volunteers, we used SVM with each annual visit (TP1, TP2, and TP3) analyzed into separate models. This separation allowed us to account for potential temporal changes in brain activity across visits.

In the subsequent analysis, within the medulloblastoma survivors' group, we conducted binary classification between high performers (standardized score ≥100) and low performers (standardized score <100) on each of seven neurocognitive tests independently: Letter-Word Identification, Reading Fluency, Passage Comprehension, Word Attack, Spelling of Sounds, Sound Awareness, and Reading Vocabulary. For this model, we focused on voxel-wise BOLD signal differences ("Letter > Color") within pre-selected ROIs and did not include additional covariates. This approach mitigates potential confounding effects of time by treating each visit separately and using BOLD signal contrasts specific to the tasks. SVM’s capability to handle high-dimensional data and its robustness to variations in input data help address the challenges posed by time effects, ensuring reliable classification performance.

Finally, regarding image representations, while the 3D template was used to provide a comprehensive overview of the regions of interest, we understand your preference for an axial presentation. Axial views are often more intuitive for readers familiar with neuroimaging studies. We have revised Figure 2 to present the results using an axial presentation, as you suggested, alongside the 3D template for comparison.

Comments 3: The results are interesting, however, the accuracy, sensitivity, and specificity for VISIT 1 are lower than other VISITS. The Discussion is in line with the results, but I suggest discussing the results of the accuracy, sensitivity, and specificity for VISIT 1. However, I suggest improving the discussion in light of previous results on Medulloblastoma patients. For this reason, I also suggest improving the bibliographic part.

Response 3: We updated the discussion, as below:

The observed improvement in SVM classification accuracy, from 57% in TP1 to 79% in TP3, reflects a progressive divergence in brain activity patterns related to orthographic processing between medulloblastoma survivors and controls over time. The observation that these neurofunctional alterations are noticed primarily at the long-term (TP3) aligns with previous studies demonstrating delayed effects of cancer treatment on brain structure and function [14,15,43-46]. The lower accuracy in TP1 may be attributed to the more subtle and less pronounced differences in brain activity shortly after treatment, when neurofunctional changes are still emerging and have not fully manifested. These early-stage changes are harder to detect, leading to reduced classification accuracy.

As time progresses, the cumulative effects of cancer treatment - particularly radiation therapy - become more pronounced, contributing to the increased accuracy observed at TP2 and TP3. Radiation therapy, a common treatment for medulloblastoma, can induce progressive alterations in brain tissue, including changes in white matter integrity and cortical thickness, which often develop gradually and may take years to fully manifest. Moreover, cognitive deficits in medulloblastoma survivors frequently emerge or worsen over time due to ongoing neuroinflammatory processes and neuroplasticity changes that continue long after treatment ends [47,48]. This gradual evolution of brain structure and function aligns with the improvement in classification accuracy seen at TP2 and TP3, when neurofunctional differences between survivors and controls are more pronounced.”

Comments 4: Moreover, I suggest adding the limitation of the study in the conclusion.

Response 4: Thank you for your suggestion. We have expanded the conclusion to include a detailed discussion of the study's limitations. The revised conclusion now addresses the following points:

“Our sample size, although representative, may limit the generalizability of our findings to broader populations of medulloblastoma survivors. We acknowledge potential confounding factors, such as socioeconomic status and pre-treatment cognitive functioning, were not accounted for in our analysis. Furthermore, while our sample is diverse, it may not capture the full range of demographic and clinical characteristics observed in the wider medulloblastoma population. The design of the fMRI RAN task could also be improved with sufficient duration of the rest condition between the letter, color and number contrasts to allow for the return of the BOLD activity to the baseline before the start of the next condition. These factors should be considered when interpreting the results and generalizing the findings.”

Round 2

Reviewer 2 Report

Comments and Suggestions for Authors

In this revised version of the study, the authors have solved all the concerns about the paper. The only thing that I can advise for the final version of the study would be about their self-critique about their sample size. The authors could test the statistical power representation of their sample size. They can find the related papers on how it can be tested or use practical guide studies about it such as given below. In my opinion, this point would improve the effect of the findings about generalization on a wider population.

Candemir, C., (2023). “A practical estimation of the required sample size in fMRI studies”, Mugla Journal of Science and Technology, 9(2), 56-63. DOI: 10.22531/muglajsci.1282492

Mumford J. , A power calculation guide for fMRI studies, SCAN (2012) 7, 738-742, doi:10.1093/scan/nss059

Author Response

Comment 1:

In this revised version of the study, the authors have solved all the concerns about the paper. The only thing that I can advise for the final version of the study would be about their self-critique about their sample size. The authors could test the statistical power representation of their sample size. They can find the related papers on how it can be tested or use practical guide studies about it such as given below. In my opinion, this point would improve the effect of the findings about generalization on a wider population.

Candemir, C., (2023). “A practical estimation of the required sample size in fMRI studies”, Mugla Journal of Science and Technology, 9(2), 56-63. DOI: 10.22531/muglajsci.1282492

Mumford J. , A power calculation guide for fMRI studies, SCAN (2012) 7, 738-742, doi:10.1093/scan/nss059

Response 1:

We appreciate the reviewer's suggestion to perform an additional test for the statistical power representation of our sample size. Unfortunately, the software tools recommended for this analysis are no longer in service:

Candemir (2023): NeuroPower - http://neuropowertools.org/neuropower/neuropowerstart/
Mumford (2012): fMRIPower - https://jeanettemumford.org/fmripower/ or https://www.nitrc.org/projects/fmripower/

As these tools are no longer available, we could not perform the specific power calculation as suggested.

Instead, we employed a Monte Carlo p-value test to assess the statistical significance of our SVM classification results. This test involved 1000 shuffles to determine the likelihood of achieving equal or higher accuracy with shuffled label data. This approach provides a robust measure of the reliability of our results and is sensitive to sample size and accuracy.

Additionally, for the univariate t-test in TP3, we used G*Power 3.19.7 software with the following parameters: alpha = 0.05 (2-tailed), 1-beta = 0.8, N2/N1 ratio = 1, Effect size = 2.95 (left) or 4.47 (right). This analysis indicated a required sample size of at least 8 (left) or 6 (right) samples, whereas our study used 42 samples.

Reviewer 3 Report

Comments and Suggestions for Authors

The manuscript was improved, and the authors addressed my concerns.

Author Response

Comment 1: The manuscript was improved, and the authors addressed my concerns.

Response 1: We thank the reviewer.